# Ultrastrong and multifunctional aerogels with hyperconnective network of composite polymeric nanofibers

Huimin He [1,5], Xi Wei[1,5], Bin Yang[1,2], Hongzhen Liu[1], Mingze Sun[1], Yanran Li[3], Aixin Yan [3], Chuyang Y. Tang [4], Yuan Lin [1,2✉] & Lizhi Xu [1,2✉]

Three-dimensional (3D) microfibrillar network represents an important structural design for various natural tissues and synthetic aerogels. Despite extensive efforts, achieving high mechanical properties for synthetic 3D microfibrillar networks remains challenging. Here, we report ultrastrong polymeric aerogels involving self-assembled 3D networks of aramid nanofiber composites. The interactions between the nanoscale constituents lead to assembled networks with high nodal connectivity and strong crosslinking between fibrils. As revealed by theoretical simulations of 3D networks, these features at fibrillar joints may lead to an enhancement of macroscopic mechanical properties by orders of magnitude even with a constant level of solid content. Indeed, the polymeric aerogels achieved both high specific tensile modulus of ~625.3 MPa cm$^3$ g$^{-1}$ and fracture energy of ~4700 J m$^{-2}$, which are advantageous for diverse structural applications. Furthermore, their simple processing techniques allow fabrication into various functional devices, such as wearable electronics, thermal stealth, and filtration membranes. The mechanistic insights and manufacturability provided by these robust microfibrillar aerogels may create further opportunities for materials design and technological innovation.

[1] Department of Mechanical Engineering, The University of Hong Kong, Hong Kong SAR, China. [2] Advanced Biomedical Instrumentation Centre Limited, Hong Kong SAR, China. [3] School of Biological Sciences, The University of Hong Kong, Hong Kong SAR, China. [4] Department of Civil Engineering, The University of Hong Kong, Hong Kong SAR, China. [5]These authors contributed equally: Huimin He, Xi Wei. ✉email: ylin@hku.hk; xulizhi@hku.hk

Porous network assembled from fibrillar elements represents an efficient structural design for materials. Nature exploited such design for the building of a variety of load-bearing biological tissues. As exemplified by the microstructures of cartilage, trabeculated bones, and plant tissues, these three-dimensional (3D) microfibrillar networks afford a combination of physical strength, lightweight, mass permeability, and surface functionality. Although extensive efforts have been devoted to the engineering of lightweight materials with 3D microfibrillar network, approaches to high mechanical strength and scalable fabrication are still limited. For example, metamaterials created from stereolithographic patterning allow rational design for desired mechanical properties[1–3], but the required structural ordering creates challenges for large-scale production. Aerogels prepared by sol-gel processes exhibit high stiffness originating from the ceramic constituents, but their intrinsic brittleness may limit their application under macroscopic deformation[4–6]. Electrospinning techniques were adopted for creating flexible polymeric membranes involving nanofibers[7,8]. However, it is difficult to design microscale topology or fabricate 3D structures with this method. Recently, aerogels or foams from self-assembled polymeric nanofibers have drawn wide attention due to their structural similarity to biological tissues[9]. Nevertheless, achieving a high mechanical strength with currently available chemistries remains difficult due to the weak interactions among fibrils[10].

Indeed, a range of biomacromolecules including cellulose[11–14] and chitin[15–17], or synthetic polymers such as polyimide (PI)[18–20] and polyurethane (PU)[21,22], were explored for the creation of nanofoams or those referred to as aerogels. Although most of them can form 3D microfibrillar network, their achievable mechanical strengths vary drastically by a few orders of magnitude (e.g., from $10^{-2}$ MPa to $10^1$ MPa) even with similar levels of solid content. It is conceivable that distinct chemical crosslinking between fibrils may affect mechanical properties of the network[23]. However, quantitative relations between the various microstructural factors and macroscopic mechanical responses remain elusive. Furthermore, the randomness of fibrillar networks creates difficulties for accurate theoretical estimation based purely on Maxwell rigidity theory[24] or Gibson-Ashby models for cellular solids[25].

Here, we report composite nanofiber aerogels (CNAs) from aramid nanofibers (ANFs) with outstanding mechanical properties. The unique interactions between nanoscale constituents lead to assembled 3D networks with high nodal connectivity and strong welded connection between fibrils. These features lead to unusually high stiffness and strength of CNAs, which is confirmed by our theoretical simulation of 3D fibrillar networks. On the other hand, successive breakage of crosslinks at high-connectivity nodes affords energy dissipation while maintaining the overall structural integrity. As a result, the fracture toughness of CNAs (~4,700 J m$^{-2}$) could be orders of magnitude higher than many existing aerogels. Furthermore, CNAs can be fabricated into various 3D configurations with simple processing steps, indicating their potential applications in wearable devices and membrane technologies. The mechanistic insights provided by these ultrastrong polymeric aerogels may create a range of opportunities for materials design and technological innovation.

## Results

**Processing and structures**. Formation of the composites involves a dispersion of ANFs in dimethyl sulfoxide (DMSO) mixed with dissolved polyvinyl alcohol (PVA). After solvent exchange with ethanol followed by critical point drying (CPD), a solid foam with nanoscale porosity will form (Fig. 1a and Supplementary Fig. 1-3), which we refer to as an aerogel for the comparison with other polymeric counterparts. We note that ANFs exhibit intrinsic branching, which is conducive to a high connectivity of the assembled network[26]. Furthermore, the extensive hydrogen bonding between the ANFs and PVA (Fig. 1b) may lead to several unique microstructural features of the resulted aerogels[27]. Firstly, the attraction between ANFs and the surrounding PVA chains may influence the topology of the assembled network during the shrinking and drying process, leading to bundling and jointing of fibrils in the 3D network (Fig. 1a and c). Such process may generate nodes connected to more than four neighboring nodes, otherwise difficult to achieve with random placement of linear fibers. Secondly, the solid PVA in the aerogel can conform to the nanofiber network, providing welded nodes with high mechanical strength (Fig. 1a). Thirdly, the reconfiguration of the hydrogen bonding may afford stress-induced orientation of nanofibers, allowing for fabrication of aerogels with desired anisotropy. All these features were observed by the scanning electron microscopy (SEM) examination. Indeed, cross-sectional images of CNAs show fibrillar networks with nodes connected to five or more neighbors (Fig. 1d), even though the 2D image of a cross section may underestimate a node's out-of-plane connection. Furthermore, the nanofibers appear to be continuous across the nodes, indicating welding by the shrunk PVA chains. Markedly, these microstructural features are also evident in a few other polymeric aerogels with high mechanical strengths, e.g., those from PI[19,20]. On the other hand, the reconfigurability of ANF-PVA interaction imparts a unique route for the anisotropic assembly of nanofibers (Fig. 1e).

The simplicity of the processing steps affords fabrication of aerogels into various configurations. The liquid precursor can be casted into 3D molds to yield bulk aerogel samples (Fig. 1f). Alternatively, it can be spin-coated or doctor-bladed to generate aerogel films (thickness of 60 μm) with semi-transparency (Fig. 1g). Ablation with common infrared lasers allows precise machining of aerogel samples at millimeter scale (Fig. 1h). These aerogels exhibit remarkable properties under loading, as exemplified by high stiffness and strength under both compression (Fig. 1i) and tension (Fig. 1j). The samples can bear mechanical loads 25,000 times (compression) and 40,000 times (tension) higher than their own weights without fracture or severe distortion.

**Mechanical properties**. Mechanical characterization quantifies the properties of CNAs with various solid contents. Porosity of the CNAs was adjusted by tuning of the concentration of their liquid precursor, with an optimized mass ratio between ANFs and PVA at 1:5 (Supplementary Fig. 4). Tensile moduli and strengths of CNAs both increase with decreasing porosity, with a high modulus of 187.6 MPa and strength of 6.3 MPa achieved at 76% of porosity, which is referred to as CNA-76 (Fig. 2a). Interestingly, the ultimate tensile strains of CNAs with various porosities remain within a relatively small range from 22.2% to 27.4%, indicating a common topology and failure mechanism for CNAs. We notice that tensile responses of existing aerogels are not widely reported due to their limited stretchability. In this regard, the high tensile modulus and strength of CNAs, in conjunction with their considerable ductility, may make them advantageous for structural applications. The compression behaviors of CNAs with various porosities follow a similar trend (Fig. 2b and Supplementary Fig. 5). However, CNA-76 exhibits a distinct linear regime at the beginning of compression with the strain below ~9%, followed by a softening stage and then ultimately strain hardening at very large compressive strain. We attribute the softening behavior to the buckling of fibrillar segments. Since CNA-76 involves a lower porosity than other samples, its fibrillar segments between nodes are shorter and can withstand a larger compressive load before buckling. In contrast, CNAs with higher

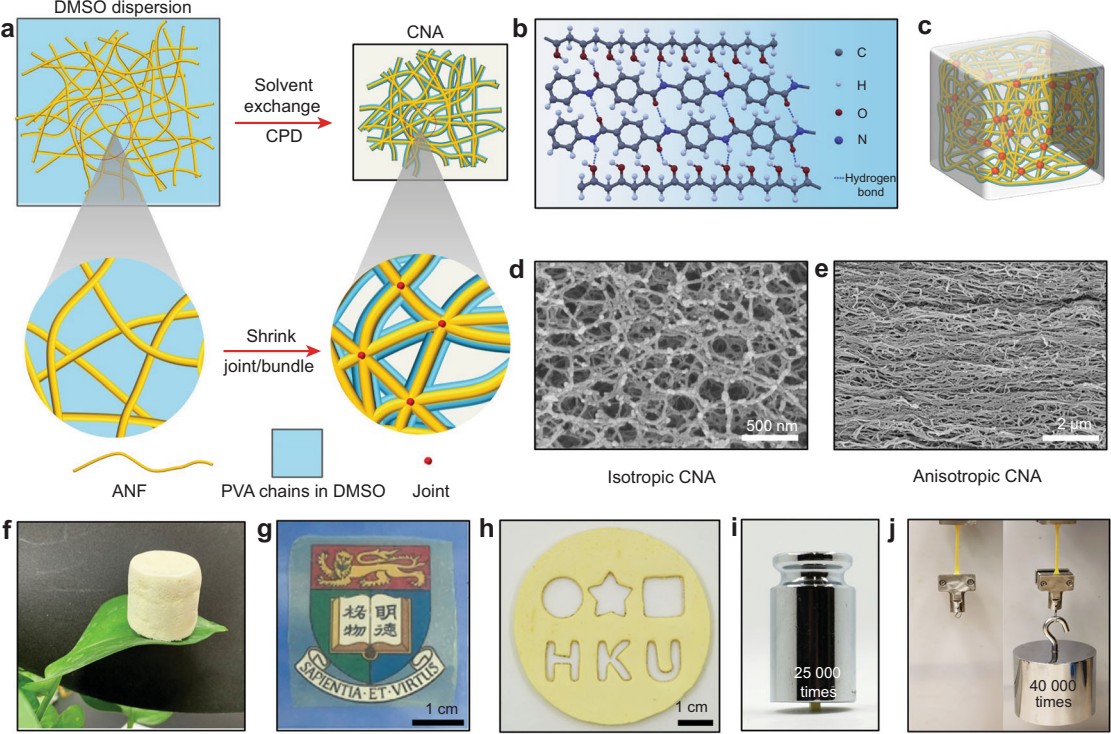

**Fig. 1 Design and architecture. a** Schematics of the assembly process of CNA. **b** A schematic of intermolecular interactions between ANF and PVA involved in CNAs. **c** A schematic of CNA involving bundling and jointing of fibrils in the 3D network. **d, e** SEM images of an isotropic CNA (**d**) and a highly oriented anisotropic CNA (**e**). **f** A photograph of a bulk CNA sample with a density of 0.02 g cm$^{-3}$. **g** A photograph of a CNA membrane with semi-transparency. **h** A CNA sample patterned with infrared laser machining. **i, j** Photographs of CNA samples under compression (**i**) and tension (**j**).

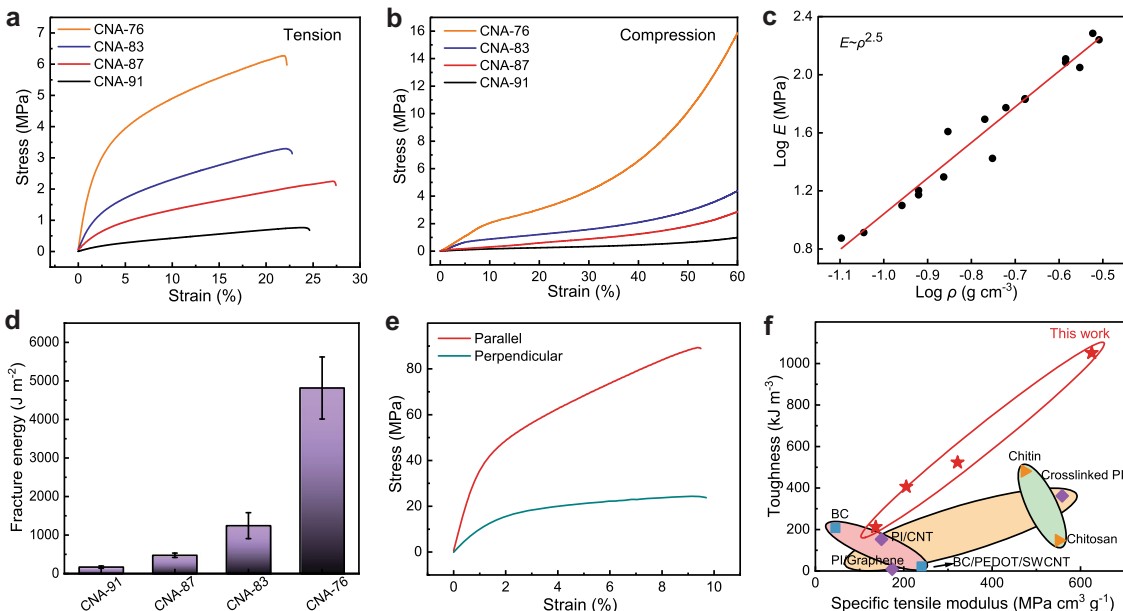

**Fig. 2 Mechanical properties. a** Tensile responses of CNAs with their porosity ranging from 76% to 91%. **b** Compressive stress-strain curves of CNAs with various porosity levels. **c** Tensile moduli of CNAs as a function of their density, obtained from experimental characterization and theoretical fitting. **d** Fracture energies of CNAs as a function of their porosity. **e** Anisotropic tensile responses of an CNA sample with highly oriented fibrils, measured from the directions parallel and perpendicular to the fibril orientation, respectively. **f** Toughness and specific tensile modulus of CNAs, as compared with other polymeric aerogels with high mechanical properties. The toughness was calculated by integrating the tensile stress-strain curves from various materials (Supplementary Table 1).

porosity may experience fibril buckling at a lower compressive load, leading to an earlier onset of softening. Finally, the strain hardening of CNAs under large compressive deformation is likely due to the densification of fibrils.

Tensile moduli of CNAs as a function of their solid contents follow a relation $E \sim \rho^{2.5}$, where $E$ and $\rho$ are the Young's modulus and density, respectively (Fig. 2c). This relation is typical for foam materials with a random internal structures[28]. Their fracture

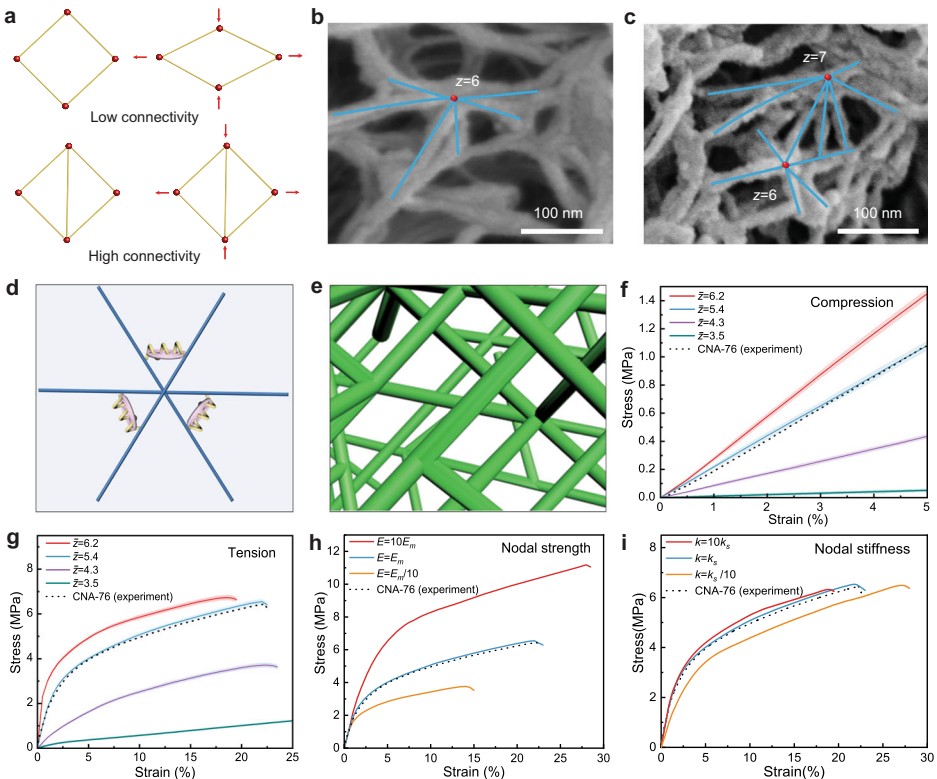

**Fig. 3 Theoretical stimulation. a** Examples of pin-jointed truss networks illustrating the distinct rigidity levels originating from different nodal connectivity. **b**, **c** SEM images of CNAs showing random 3D network with high-connectivity nodes. **d** A schematic model with linear and rotational springs connecting each pair of crosslinked fibrils. If a welded node contains three crosslinked fibrils, 3 pairs of linear and rotational springs will be introduced to represent the interconnection among them. **e** A 3D network representing CNA with an average nodal connectivity of $\bar{z} = 5.4$. **f**, **g** Effects of average nodal connectivity on the compressive (**f**) and tensile (**g**) responses of the simulated random 3D networks. Results from the simulation of $\bar{z} = 5.4$ match the experimental observations from CNA-76. **h**, **i** Effects of nodal strength represented by the binding energy, $E$ (**h**), and nodal stiffness represented by the spring constant, $k_s$ (**i**), of the crosslinkers, on the tensile responses of simulated 3D networks with an average nodal connectivity of $\bar{z} = 5.4$.

energy also increases with increasing solid content, following a typical trend for foams (Fig. 2d, Supplementary Fig. 6 and 7). We note that the high fracture energy of CNA-76 (~4697.6 J m$^{-2}$) exceeds those of many engineering polymers and is actually comparable to the fracture energies of natural rubbers[29]. The toughness of CNAs may arise from successive breakage of connection between fibrils, as well as their re-orientation and relative sliding under imposed deformation[27]. Indeed, the characteristics of isotropic CNAs are distinguishable from those of other polymeric aerogels known for their high mechanical properties (Fig. 2f). Furthermore, anisotropic porous composites can be obtained from stress-induced alignment of the nanofiber network during processing. For samples with a porosity of 48.7%, the tensile modulus and tensile strength along the orientation of the fibrils can reach 4.3 GPa and 89.3 MPa, respectively (Fig. 1e and Fig. 2e). These parameters are ~4.2 and 3.7 times higher than the properties measured along the directions perpendicular to the aligned fibrils.

**Theoretical modeling.** A unique feature of CNAs is that many fibrils are connected and welded at their common joints (Fig. 3b and c), leading to a high nodal connectivity in contrast to networks from random placement of linear fibrils. We proceed by investigating how the connectivity and nodal mechanics of random 3D networks influence the macroscopic properties of the material. To illustrate the physical pictures, it is worthwhile to review some of the established models for the mechanics of network structures. In Maxwell rigidity theory, a pin-jointed truss network will exhibit a finite stiffness only if its connectivity

reaches a threshold value of $2d$, with $d$ being the dimension of the problem[30]. Below the threshold, free rotation of the trusses will accommodate the deformation, leading to a zero network stiffness (Fig. 3a). On the other hand, once the connectivity exceeds the critical value, the network behavior changes from fluid- to solid-like. Recent works showed that the threshold connectivity will decrease if bending rigidity of individual fibrils is considered[30]. These models highlight the significance of nodal connectivity for the mechanics of fibrillar networks. However, they could not replicate the observed response of CNAs (Supplementary Fig. 8) because several important factors were not fully considered. Firstly, fibrillar joints may exhibit resistance to rotation, providing a significant contribution to the overall rigidity. Secondly, the breakage of fibrillar joints should play a major role in the mechanical responses under high deformation, which was not included in conventional models. Thirdly, the random nature of fibrils and nodes in the 3D network requires additional consideration in the modeling (Fig. 3b, c).

We developed a computational model to address these issues. Specifically, an ordered fibril network was constructed according to a face-centered-cubic (FCC) lattice with a high initial connectivity of $z = 12$[30]. Random deletion of fibrillar segments were implemented to reduce the connectivity of the network to desired levels (Fig. 3e). Stochastic displacements were imposed to each node to increase the randomness of the network. At the fibrillar joints, combined linear and angular springs, i.e., crosslinkers, were added between each pair of fibrils to restrain their relative separation and rotation (Fig. 3d). In addition, breakage of the crosslinker was assumed to take place once the stored elastic energy exceeds a

critical value (Supplementary Fig. 9), defined as the binding energy of the crosslinker[31]. Note that, physically, this binding energy can be attributed to the interactions between ANF-PVA, ANF-ANF, and PVA-PVA at the fibrillar joints. Finite element method was used to simulate the response of the network under imposed strain with periodic boundary conditions. Individual fibrils were modeled as 3D nonlinear Reissner beams that can undergo large stretching, bending, and twisting deformation[30]. The Young's modulus and diameter of fibrils are set to be $E = 18.2$ GPa and $d = 50$ nm, respectively, emulating the mechanics of fibrils in CNAs. The lengths of fibrils (or equivalently the mean distance $l_c$ between two nodal points, refer to Supplementary Table 2) in each network were adjusted by a scaling factor to ensure a constant solid fraction of ~24%, corresponding to that of CNA-76. The elongation of CNAs is mostly governed by the bending, enforced deformation and breakage of crosslinkers at nodal points, rather than the stretching and fracture of individual fibrils (Supplementary Fig. 10).

We constructed groups of 3D networks with average connectivity of $\bar{z} = 3.5$, 4.3, 5.4, and 6.2 (Supplementary Fig. 11) and simulated their responses under imposed deformation. Interestingly, the compressive stiffness of the networks increases drastically with increasing nodal connectivity, despite that all the networks are formed from the same material with a constant solid fraction (Fig. 3f). This observation highlights the significance of nodal connectivity in CNAs. We note that all the simulated networks exhibit a finite rigidity even with an average connectivity below the rigidity threshold in Maxwell model ($z_{cf} = 6$). This behavior correlates to the fact that jointed fibrils cannot freely rotate with respect to each other, allowing networks with a low connectivity to resist deformation.

The tensile behaviors of the networks were simulated with a linearly increasing imposed strain until their total fracture. Similar to the behaviors under compression, tensile moduli of the networks exhibit a strong dependence on their average nodal connectivity (Fig. 3g). Furthermore, after an initial linear stage, the networks experience significant strain softening, which agrees with the experimental observation. Further examination shows that such softening behavior is caused by the successive breakage of crosslinkers (Supplementary Fig. 12), which indicates that networks with high connectivity allow energy dissipation while maintaining structural integrity. In this regard, the binding energy of crosslinkers should play a critical role in determining the macroscopic strength and toughness of the network. Indeed, increasing the binding energy leads to an elevated fracture strength of the material (Fig. 3h). In contrast, a lower binding energy of the crosslinkers will trigger an earlier softening of the material and a reduced fracture strength. These simulation results highlight the significance of welded high-connectivity joints in CNAs for their high tensile strength and toughness. On the other hand, changing the stiffness (i.e., the effective spring constant) of crosslinkers has only a minor influence on the macroscopic properties of the network (Fig. 3i and Supplementary Fig. 13). The high energy dissipation capability of CNAs can also be seen from their hysteresis under cyclic tensile loading (Supplementary Fig.14 and 15). Finally, we note that strain-stiffening commonly observed in water-rich biological tissues did not occur in our simulation[32,33], which is reasonable due the presence of successive breakage of fibrillar joints within aerogel materials.

**Technological applications.** In addition to the superb mechanical properties, the manufacturability of CNAs enables a range of applications as membrane devices. For instance, the porosity of CNAs enables selective mass transport and air filtration (Fig. 4a and Supplementary Fig. 16). The average pore sizes of CNAs range from 140 nm to 1,463 nm, which can be adjusted by tuning

of the solid content (Fig. 4b). CNA membranes with a thickness of ~20 μm exhibited a good air permeability, with a pressure drop of 0.9-2.2 kPa under a face velocity of 0.05 m s$^{-1}$ (Fig. 4c). They showed excellent bacterial filtration efficiency on par with commercial membranes based on mixed cellulose ester (MCE) (Fig. 4d), while the fracture toughness of CNAs can be an order of magnitude higher than that of MCE (Supplementary Fig. 17). In another application, the CNA membranes can be utilized for wearable electronics. Conductive inks can infiltrate into the porous structures of CNAs to generate patterns of electrodes and interconnects (Fig. 4e). Kirigami structures can be introduced into the CNA membrane, leading to a reconfigurable device conformal to the 3D surface of skin (Fig. 4f, g). In this case, the high fracture toughness of CNAs is beneficial for preventing cracking at the edges of kirigami cuts. Furthermore, the low thermal conductivity and high infrared absorption of CNA membranes may enable thermal stealth in a wearable configuration. In particular, the thermal conductivity of CNAs drops from ~0.028 to 0.014 W m$^{-1}$ K$^{-1}$, as the porosity increases from 76% to 91% (Fig. 4h). In addition, CNAs exhibit a strong absorption of mid-infrared (IR) radiation at 9.3 μm, which is close to the peak emission wavelength (~9.5 μm) by the human body (Fig.4i)[34]. To demonstrate thermal stealth behaviors, two membrane rings from silicone and CNA were put on the index and middle fingers of a volunteer, respectively (Fig. 4j). The contrast in thermal images confirms the thermal stealth property of CNAs as compared with silicone.

## Discussion

In summary, we have developed a class of polymeric aerogels with outstanding mechanical properties originating from their hyper-connective fibrillar network. The theoretical models developed in this work accurately depicted the observed mechanical behaviors of CNAs, revealing quantitative relationships between configurations of fibrillar joints and macroscopic material properties. The mechanistic insights obtained from these models should be applicable to the engineering of a range of porous materials involving fibrillar networks. Further analysis would benefit from advanced characterization techniques revealing 3D topological details of microfibrillar networks, as well as sophisticated models including additional microstructural factors. From technological perspectives, the excellent mechanics, porosity, and manufacturability of these polymeric aerogels may create diverse opportunities for flexible electronics, energy systems, biomedical devices, and other applications.

## Methods

**Preparation of CNAs.** Kevlar para-aramid pulp (Type 979; DuPont) and poly (vinyl alcohol) (PVA; Mw: $146,000 − 186,000$; 99%+ hydrolyzed; Sigma-Aldrich) were used for the preparation of liquid precursors for CNAs. Briefly, Kevlar pulp was dispersed in DMSO (3 wt%) under magnetic stirring at 95 °C for 7 days. PVA was dissolved in DMSO (15 wt%) under magnetic stirring at 95 °C for 7 days. Mixing these two liquid precursors with 1:1 mass ratio leads to the precursor of CNA-76. Precursors of CNAs with various porosities were obtained by adding additional pure DMSO to the mixture, retaining the mass ratio between dispersed ANFs and dissolved PVA at 1:5. For the preparation of bulk or film samples, the liquid mixture of ANF-PVA was poured into a mold or casted on a flat aluminum foil using a film coater. Solidification of ANF-PVA mixtures was achieved by solvent exchange in deionized (DI) water, leading to solid hydrogels. The hydrogel samples were immersed in ethanol for another 24 h followed by critical point drying (CPD, Tousimis Autosamdri 931) to generate CNAs.

**Structural characterization of CNAs.** The shrinkage of aerogels during drying was determined by:

$$\text{Shrinkage} = \left(1 - \frac{V_a}{V_h}\right) \times 100\% \tag{1}$$

where $V_h$ and $V_a$ are the volumes of samples before and after supercritical $CO_2$ drying, respectively.

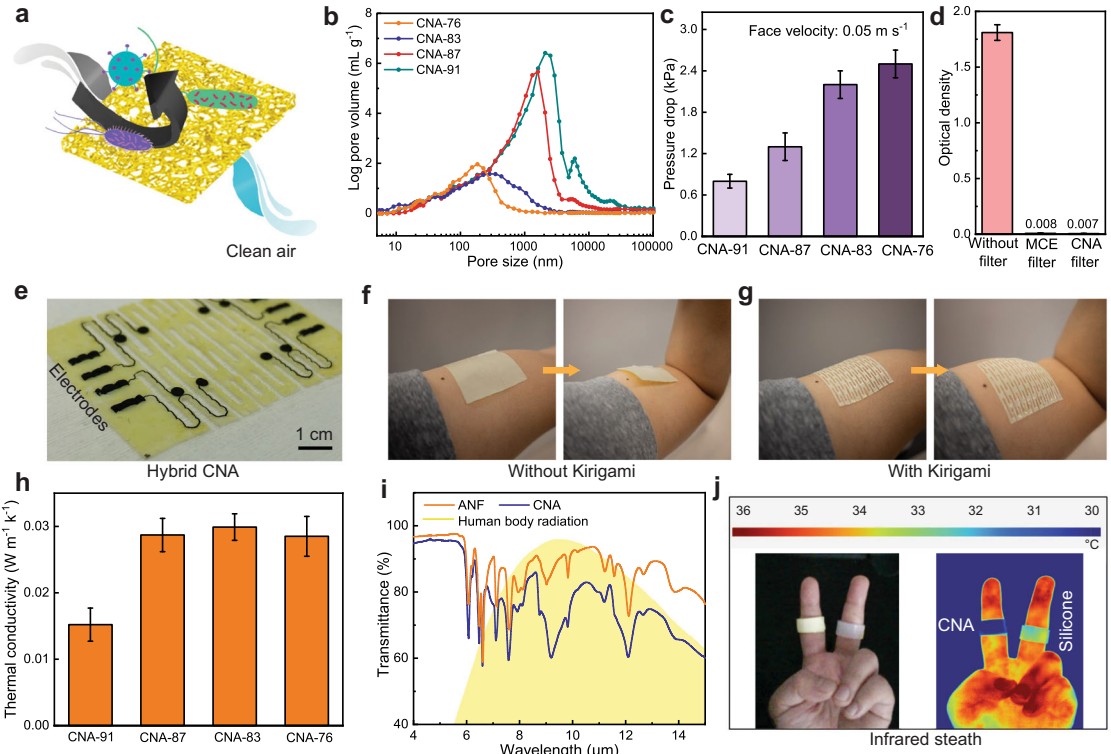

**Fig. 4 Potential applications. a** A schematic of a CNA membrane serving as an air filter. **b** Pore size distribution of CNAs with various levels of porosity. **c** The drop of air pressure across 20-μm-thick CNA membranes, measured under a face velocity of 0.05 m s⁻¹. **d** Filtration efficiency of CNA membranes against airborne bacteria, as compared with commercial MCE membranes. The optical density at 600 nm wavelength ($OD_{600}$) is proportional to cell concentration. **e** A kirigami CNA membrane with a pattern of conductive ink infiltrated into the porous structures, serving as electrodes and interconnects for wearable electronics. **f, g** Conformability of a kirigami CNA membrane on the dynamic 3D surface of human skin (**g**), as compared with a membrane without kirigami (**f**). **h** Thermal conductivities of CNAs with various levels of porosity. **i** Infrared transmittance spectra of CNA and pure ANF, as compared with the spectrum of human body radiation. **j** A photograph and an infrared thermal image showing the thermal stealth behavior of a CNA membrane as compared with silicone.

Morphology of aerogels was observed under scanning electron microscope (SEM; Hitachi S4800 FEG). To identify the inner network of aerogels, hydrogel samples were frozen by liquid nitrogen and cut into halves to expose cross-sections and then dried by supercritical $CO_2$.

The porosity of CNAs was calculated by:

$$Porosity = \left(1 - \frac{6\rho_b}{\rho_a + 5\rho_p}\right) \times 100\% \qquad (2)$$

where $\rho_b$ is the density of bulk CNA sample. The coefficients in the equation are based on the mixing ratio between ANF and PVA (1:5). The densities of ANF ($\rho_a = 1.44$ g cm⁻³) and PVA ($\rho_p = 1.19$ g cm⁻³) are provided by suppliers. The pore size distribution of CNAs with different densities was measured by mercury intrusion porosimeter (MIP, AutoPore IV 9600) with pressure ranging from 0.1 to $6.1 \times 10^4$ psi.

**Mechanical tests.** For tensile tests, samples were cut into dumbbell shape (15 mm in length, 3 mm in width and 1.5 mm in thickness) and loaded with a tensile-compressive tester (Zwick Roell) at a deformation rate of 100% min⁻¹. Cylinder samples (10 mm in diameter and 3 mm in thickness) were tested in compression with a deformation rate of 50% min⁻¹.

For a comparison with data reported in literature, we calculated the area under the stress-strain curve until fracture as an estimate of the toughness for various materials, with a unit of kJ m⁻³. Fracture energy ($\Gamma$) of CNAs was measured by both tearing test and pure shear test[35], with a unit of J m⁻². For the tearing test, film samples (8 mm in width, 50 mm in length and 300 μm in thickness) were cut into a trouser shape with a notch of 20 mm in length. The two arms of samples were mounted on the mechanical tester and stretched at a constant velocity of 1.7 mm s⁻¹. The $\Gamma$ was calculated from the steady state tearing force $F$ as

$$\Gamma = 2F/t, \qquad (3)$$

where $t$ is the thickness of the sample.

Rectangular samples (30 mm in width ($w$), 20 mm in length and 0.8 mm in thickness ($t$)) were used for the pure shear test. A 15-mm notch was cut into the width of the samples. Samples were mounted on two clamps with a fixed distance

of 15 mm. The force-extension curves were recorded at an extension rate of 1.7 mm s⁻¹. The fracture energy was calculated by:

$$\Gamma = U(L_C)/(w \times t) \qquad (4)$$

where $U(L_C)$ is the work done by the tensile force on a unnotched samples up to a critical extension length ($L_C$), which is determined by the extension distance when crack starts to propagate for the corresponding notched samples. The fracture energies obtained from tearing tests and pure shear tests are consistent (Supplementary Fig. 6 and 7).

**Modeling the deformation of microfibrillar networks.** 3D fibrillar networks interconnected by breakable crosslinkers were generated in silico from FCC lattice (with dimension $6\sqrt{2}l_c \times 6\sqrt{2}l_c \times 6\sqrt{2}l_c$ where $l_c$ is the distance between two neighboring nodes) as proposed by Broedersz et al.[30]. Notice that perfect FCC lattice networks have an initial coordination number $z = 12$, placing them well above the central-force (CF) isostatic threshold connectivity($z_{cf} = 6$) according to the Maxwell rigidity theory. In contrast, most previous models for stiff-fiber networks have a maximum coordination number 4. Here, we explore the effects of network connectivity by randomly removing fibrillar segments between nodes until the desired connectivity is reached. At the same time, the value of $l_c$ is chosen to ensure the fiber density equals to that in our experiment. Six random networks were generated at each connectivity level to make sure results and conclusions obtained from the simulations are representative enough. Finite element method was then used to capture the deformation responses of networks where each fibril was modeled as a three-dimensional Reissner beam[36] that can undergo large rotation, stretching, bending, and twisting. The Young's modulus of fibrils was taken to be $E = 18.2$ GPa, given that the mass ratio between ANF and PVA is 1:5 and the tensile moduli of PVA and Kevlar 979 fiber are ~960 MPa (measured experimentally) and ~ 123 GPa[37], respectively. The diameter of fibrils was taken to be $d = 50$ nm, which corresponds to the approximate diameter observed in SEM[27]. Each pair of jointing fibrils were assumed to be connected by crosslinkers modeled by linear and rotational springs with spring constants of $\kappa_s$ and $\kappa_r$ respectively.

Consequently, the strain energy stored in each deformed crosslinker is

$$E_c = \frac{1}{2}\kappa_s \delta l^2 + \frac{1}{2}\kappa_r \delta\theta^2 \qquad (5)$$

where $\delta l$ is the separation of the intersecting points and $\delta\theta$ is the change in the relative angle between two fibrils. In addition, once the energy reaches a critical value $E_m$, the crosslinker will break. Finally, periodic boundary conditions were enforced on each side of the simulation box. During simulation, the bottom side of the box was assumed to be fixed when the top side was forced to move in the vertical direction according to how the tensile or compression strain was applied to the material in our experiments, and the reaction force was recorded, allowing us to construct the stress-strain curve. Unless stated otherwise, the values of parameters adopted in our simulations are listed in Supplementary Table 2.

**Air filtration tests**. CNA-83 films with a thickness of 20 μm were tested and compared with commercial MCE films (MF-Millipore™ Membrane Filter, 0.22 μm pore size). Briefly, ambient air was pumped through filtration funnels (Corning) for 4 days, with a polyethersulfone (PES) film collecting residual airborne bacteria after the filtration (Supplementary Fig. 16). The air flow rate was controlled at 2 L min$^{-1}$. After 4 days of air filtration, PES collectors were immersed in 10 ml of Lysogeny broth (LB) in 50 ml Falcon tubes. Following shaking incubation at 37 °C with 220 rpm agitation for 24 h, the concentration of bacteria in the suspension was quantified by measuring the optical density at 600 nm (OD$_{600}$) with a spectrophotometer (Ultrospec 2100 pro). Data is expressed as the mean of three replicates. The permeability of CNA membranes was also tested with additional experiments (Supplementary Figs. 18 and 19).

**Characterization of thermal properties**. CNA samples with 100 mm in diameter and 1 mm in thickness were measured with thermal conductivity analyser (Hot Disk TPS 3500). The IR transmittance was measured from aerogel films with a thickness of 200 μm, using a Fourier-transform infrared (FTIR) spectrometer (Thermofisher IS50). Thermal images were taken with an infrared imaging camera (Fluke Ti480).

## Data availability
The data supporting the findings of this study are available within the article and its Supplementary Information as well as Source Data. Source data are provided with this paper.

## Code availability
The custom Matlab code for generating the 3D fiber networks reported in the paper has been deposited to Source Data. Source Data are provided with this paper.

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

## Acknowledgements
The study is supported by Research Grants Council (RGC), University Grants Committee (UGC) (Project 27210019 and 17200320 to L.X.; 17210618 and 17210520 to Y.L.). The study is also supported by the Health and Medical Research Fund (HMRF), Food and Health Bureau of Hong Kong SAR (Project 18171042 to A.Y.). The authors thank Hao Li, Wenxiu Li, Wei Xu and Yi Zhang for experimental assistance and fruitful discussions.

## Author contributions
L.X. designed and supervised the research. H.H. carried out experimental investigations. Y.Lin supervised theoretical simulations. X.W. carried out theoretical simulations. B.Y., H.L., M.S., Y.Li, C.Y.T. and A.Y. contributed to the experiments and analysis. L.X., Y.Lin, H.H., X.W. and B.Y. co-wrote the manuscript.

## Competing interests

The authors declare no competing interests.
