## [Peer Review File · Nature Communications]

Ultrastrong and Multifunctional Aerogels with
Hyperconnective Network of Composite Polymeric NanofibersReviewers' Comments:

Reviewer #1:

Remarks to the Author:

Review of "Ultrastrong and Multifunctional Aerogels with Hyperconnective Network of Composite Polymeric Nanofibers" by Xu et al.

The authors developed a new class of aerogels composed of self-assembled 3D aramid nanofiber networks and polyvinyl alcohol (PVA) through simple processing techniques. By tuning the concentrations and material ratios in the fabrication process, the authors achieved outstanding macroscopic mechanical properties, e.g., light-weight, high strength and high fracture toughness, which outperform the existing polymeric aerogels. More impressively, the authors developed and extended the Maxwell rigidity theory of pin-jointed truss network by including the rotation, breakage and random nature of fibrillar joints which can capture/predict the main features of the mechanical responses of such microfibrillar networks, illustrating the roles of the fibrillar joints. In the end, the authors demonstrated potential applications of the developed aerogels in wearable electronics, thermal stealth, and filtration membranes.

The results of the paper are interesting and encouraging in that they show the potential for the large-scale production of aerogels with superior mechanical properties and multifunctionalities. The theoretical model revealed that the fibrillar joints ("crosslinking") play key roles in the enhancement of mechanical properties. This reviewer thinks this paper constitutes a significant contribution to basic science as well as potential technological applications, and therefore would like to recommend publication if the following questions can be addressed.

1) In the abstract, the claim "the polymeric aerogels achieved both high tensile strength of ~ 89 MPa and fracture energy of $\sim 4,700$ J/m²" sounds like that the aerogels can achieve the two properties simultaneously. However, the tensile strength (~ 89 MPa) was obtained in an anisotropic sample in the parallel direction while the fracture energy was obtained in the isotropic sample (CNA-76). The authors should avoid potential misleading sentences.

2) The modeling shows that the fibrillar joints play critical roles in the mechanical responses, which are simplified as springs ("crosslinkers") between each pair of fibers. Can the authors discuss more about the underlying physics of such "crosslinkers"? Does the "binding energy" of the "crosslinker" originate from the cohesive fracture energy of the welding PVA between fibers, or interactions between fiber and PVA or even fiber and fiber?

3) Current modeling treats nodes as independent pairs of springs that fail independently. However, it is not unlikely that these springs might be correlated (e.g., node is formed/stabilized through fixation of PVA wrappings). How does such a scenario affect the overall mechanical responses? What if a different failure criteria of node (e.g., node fails when a single spring fail or the total strain energy reach a critical value) is adopted?

4) Does the porosity affect nodal connectivity of the 3D network in experiments?

5) The authors indicate that ANFs exhibit intrinsic branching. How does such "intrinsic branching node" affect the overall nodal connectivity and nodal properties of the resultant 3D networks?

Minor comments:

In Supplementary Table 1, it seems that "CNA-89" should be "CNA-91" or "CNA-91" is missing?

Reviewer #2:

Remarks to the Author:

The manuscript, "Ultrastrong and Multifunctional Aerogels with Hyperconnective Network of Composite Polymeric Nanofibers" by Lin, Xu and co-workers present an aerogel material made of fibrils that have meta-aramid (Kevlar) core / PVA shell structure. The interesting class of aerogel material was previously shown to have superior mechanical property from the corresponding author (L. Xu)'s 2018 paper (Ref [27] in the manuscript). The novelty of the current manuscript lies in its explanation of the superior mechanical property (mostly related with its stretchability) based on simulation model based on 3D nonlinear Reissner beams with multiple connectivity. In addition, the manuscript includes proof-of-concept demonstrations of potential applications, including air filtration, wearable electronics with kirigami structure, and thermal insulator. In the reviewer's view, the study is potentially very interesting for interdisciplinary audiences, but the novelty aspect of the manuscript can be justified if there is enough evidence to justify the efficacy of the simulation to explain the experimental results. The current manuscript, though, needs to improve in terms of such justifications. Specifically, more details must be revealed both from the experimental and simulation sides, and the simulation must be better justified. Please see below.

(1) The most critical problem of the manuscript is that the information about the 'fiber' is very scarce so that its not possible for the reviewer to judge whether the use of Reissner beam model can be justified. Specifically,

(1-1) For an 'average' fiber, what is the diameter of the Kevlar core and the thickness of the PVA sheath? The authors described an overall diameter of 50 nm, their ratios to be 1:5 based on mass ratio, and their moduli to be ~960 MPa. But I question about these assumptions based on my points (1-2) and (1-3).

(1-2) Mechanical property of PVA sheath changes drastically with the water content inside of the PVA. Was the water content controlled? If so, how?

(1-3) One way to justify the use of Reissner beam may be an experimental tensile test of 'a model core/shell fiber'. Here, my biggest concern is that we do not know the origin of the seemingly high stretchability/compressibility of the aerogel. While Kevlar is supposed to fail at low strain, is it only PVA that sustain the overall structure of the fibers? If so, when do we see such failure?

(2) Related with the point (1), I suggest that a multiple 'hysteresis' strain cycles must be performed to supplement Figures 2a,2e, and 3g.

(3) The description about the simulation model is also thin in the current manuscript. What does the cartoon in Figure 3d mean? I see springs and an indication that they can rotate. But what does that really mean, especially in terms of the overall network structure (Fig 3e)? And how were the "average connectivity" shown in Figure S10 calculated for all stress/strain datapoints? How does 'buckling', 'twisting', and 'stretching' taken into account in the Reissner beam model that the authors used? The explanations given in the current manuscript is not enough for readers to figure out the boundary conditions that was actually used in the manuscript; especially in terms of their relations with the actual experimental conditions.

(4) Minor points:

(4-1) "The average pore sizes of CNAs range from 140 nm to 1463 nm..." ==> How were these values quantified?

(4-2) "For anisotropic CNAs with a porosity of 48.7%" ==> Porosity seems to be quite low. Can this be classified as 'aerogel'?

Comments from Reviewer #1

Summary: Review of “Ultrastrong and Multifunctional Aerogels with Hyperconnective Network of Composite Polymeric Nanofibers” by Xu et al.

The authors developed a new class of aerogels composed of self-assembled 3D aramid nanofiber networks and polyvinyl alcohol (PVA) through simple processing techniques. By tuning the concentrations and material ratios in the fabrication process, the authors achieved outstanding macroscopic mechanical properties, e.g., light-weight, high strength and high fracture toughness, which outperform the existing polymeric aerogels. More impressively, the authors developed and extended the Maxwell rigidity theory of pin-jointed truss network by including the rotation, breakage and random nature of fibrillar joints which can capture/predict the main features of the mechanical responses of such microfibrillar networks, illustrating the roles of the fibrillar joints. In the end, the authors demonstrated potential applications of the developed aerogels in wearable electronics, thermal stealth, and filtration membranes.

The results of the paper are interesting and encouraging in that they show the potential for the large-scale production of aerogels with superior mechanical properties and multifunctionalities. The theoretical model revealed that the fibrillar joints (“crosslinking”) play key roles in the enhancement of mechanical properties. This reviewer thinks this paper constitutes a significant contribution to basic science as well as potential technological applications, and therefore would like to recommend publication if the following questions can be addressed.

Our Response: We thank the Reviewer for the enthusiastic comments and support for the publication of our work in *Nature Communications*.

Comment #1: In the abstract, the claim ‘the polymeric aerogels achieved both high tensile strength of ~89 MPa and fracture energy of ~4,700 J/m²’ sounds like that the aerogels can achieve the two properties simultaneously. However, the tensile strength (~89MPa) was obtained in an anisotropic sample in the parallel direction while the fracture energy was obtained in the isotropic sample (CNA-76). The authors should avoid potential misleading sentences.

Our response: We are grateful to the Reviewer for bringing this issue to our attention. Following the reviewer’s suggestion, the sentence in the abstract has been modified, reflecting the properties obtained from CNA-76.

Our modification to the manuscript: We changed the related description in the abstract to: “the polymeric aerogels achieved both high specific tensile modulus of ~625.3 MPa·cm³/g and fracture energy of ~4,700 J/m²”

Comment #2: The modeling shows that the fibrillar joints play critical roles in the mechanical responses, which are simplified as springs (“crosslinkers”) between each pair of fibers. Can the authors discuss more about the underlying physics of such “crosslinkers”? Does the “binding energy” of the “crosslinker” originate from the cohesive fracture energy of the welding PVA between fibers, or interactions between fiber and PVA or even fiber and fiber?

Our response: We thank the Reviewer for this suggestion. We revised the manuscript to include a brief discussion on the physical origins of the “binding energy” of crosslinkers introduced in our model.

Our modification to the manuscript: On page 9, paragraph 1, we added “Note that, physically, this binding energy can be attributed to the interactions between ANF-PVA, ANF-ANF, and PVA-PVA at the fibrillar joints.”

Comment #3: Current modeling treats nodes as independent pairs of springs that fail independently. However, it is not unlikely that these springs might be correlated (e.g., node is formed/stabilized through fixation of PVA wrappings). How does such a scenario affect the overall mechanical responses? What if a different failure criteria of node (e.g., node fails when a single spring fail or the total strain energy reach a critical value) is adopted?

Our response: The referee has raised a very important point. Indeed, an independent pair of springs were used in our original simulation to restrain the relative separation and rotation of each pair of fibrils connected at the node. In reality, as correctly pointed out by the reviewer, it is conceivable that these springs may be correlated, i.e. they may fail at the same time rather than independently. To assess the influence of such possibility on the overall response of the material, we have conducted additional simulations based on different failure criteria of crosslinkers. Specifically, for a welding point containing N_f fibers, there are $N_c = \frac{N_f(N_f-1)}{2}$ crosslinkers interconnecting different fibrils. If we assume these N_c crosslinkers fail simultaneously once the total strain energy stored in them reaches $N_c E_c$ (with E_c being the binding energy of each crosslinker defined in our original model), the simulated tensile response of CNA gel is shown by the blue line in the newly added Supplementary Fig. 9. Clearly, under such circumstance, the stress and failure strain all become slightly higher than those when crosslinkers were allowed to break independently (i.e. when the strain energy stored in any on them is above E_c). This is not surprising because some crosslinkers will undergo larger distortion than others and therefore will break first, which weakens the whole material and eventually leads to a lower stress level and fracture strain. On the other hand, if we assume that the linear and rotational springs in each crosslinker can rupture independently (i.e. once the energy stored in them reaches $E_c/2$), then the material will rupture at an even lower stain level (see the red line in Supplementary Fig. 9). Further examination of our simulation results revealed that the strain energy stored in the linear spring in most crosslinkers is higher than that in the rotational spring. Consequently, many liner springs will break earlier under this new criterion, ultimately resulting in the earlier fracture of the whole material.

Nevertheless, it can be seen that the overall shape/trend of the stress-strain curve under different breakage criteria of crosslinkers remains largely the same, indicating the main physics should have been captured by our model.

Our modification to the manuscript: We added Supplementary Fig. 9 to show how different failure criteria of crosslinkers influence the overall tensile response of the network.

Supplementary Fig. 9 **Stress-strain curves of CNA-76 under different crosslinker failure criteria.** For a welded node containing N_f fibers, there are $N_c = \frac{N_f(N_f-1)}{2}$

crosslinkers interconnecting different fibers. If we assume these N_c crosslinkers fail simultaneously once the total strain energy stored in them reaches $N_c E_c$ (with E_c being the average binding energy of each crosslinker), the simulated tensile response of CNA is shown by the blue line. Clearly, under such circumstance, the stress and failure strain all become slightly higher than those when crosslinkers were allowed to break independently (i.e. when the strain energy stored in any on them is above E_c). This is not surprising because some crosslinkers will undergo larger distortion than others and therefore will break first, which weakens the whole material and eventually leads to a lower stress level and fracture strain. On the other hand, if we assume that the linear and rotational springs in each crosslinker can rupture independently (i.e. once the energy stored in them reaches $E_c/2$), then the material will rupture at an even lower strain level (red curve). Further examination of our simulation results revealed that the strain energy stored in the linear spring in most crosslinkers is higher than that in the rotational spring. Consequently, many liner springs will break earlier under this new criterion, ultimately resulting in the earlier fracture of the whole material. Nevertheless, it can be seen that the overall shape/trend of the stress-strain curve under different breakage criteria of crosslinkers remains largely the same, indicating the main physics should have been captured by the model.

Comment #4: Does the porosity affect nodal connectivity of the 3D network in experiments?

Our response: Directly from SEM images (Supplementary Fig. 2) for CNAs with various porosities, negligible differences in the nodal connectivity were observed. The connectivity and the network topology for various CNAs appear to be consistent implied by mechanical cues as well: (1) The failure strains under tension stay within a relatively narrow range (Fig. 2a). (2) The modulus-density relationship for CNAs follows a typical scaling law: $E \sim \rho^{2.5}$, indicating consistent network topology. Detailed analysis on topology-porosity relationship is subject to future work and will be reported separately.

Comment #5: The authors indicate that ANFs exhibit intrinsic branching. How does such “intrinsic branching node” affect the overall nodal connectivity and nodal properties of the resultant 3D networks?

Our response: The observed branching of ANFs is conducive to forming highly interconnected network as compared to purely linear fibrils. However, the high nodal connectivity of CNAs can be related to additional factors such as jointing and bundling of fibrils. It is difficult to independently quantify the contribution of these factors through experiments. We will research into additional approaches to the investigation of network topography and present our findings in future reports.

Minor comments: In Supplementary Table 1, it seems that “CNA-89” should be “CNA-91” or “CNA-91” is missing?

Our response and modification to the manuscript: We want to thank the reviewer for bringing these typos/errors to our attention. They have all been corrected in the revised submission.

Comments from Reviewer #2

Summary: The manuscript, "Ultrastrong and Multifunctional Aerogels with Hyperconnective Network of Composite Polymeric Nanofibers" by Lin, Xu and co-workers present an aerogel material made of fibrils that have meta-aramid (Kevlar) core / PVA shell structure. The interesting class of aerogel material was previously shown to have superior mechanical property from the corresponding author (L. Xu)'s 2018 paper (Ref [27] in the manuscript). The novelty of the current manuscript lies in its explanation of the superior mechanical property (mostly related with its stretchability) based on simulation model based on 3D nonlinear Reissner beams with multiple connectivity. In addition, the manuscript includes proof-of-concept demonstrations of potential applications, including air filtration, wearable electronics with kirigami structure, and thermal insulator. In the reviewer's view, the study is potentially very interesting for interdisciplinary audiences, but the novelty aspect of the manuscript can be justified if there is enough evidence to justify the efficacy of the simulation to explain the experimental results. The current manuscript, though, needs to improve in terms of such justifications. Specifically, more details must be revealed both from the experimental and simulation sides, and the simulation must be better justified. Please see below.

Our Response: We thank the Reviewer for the comments and finding our work interesting.

Comment #1: (1) The most critical problem of the manuscript is that the information about the 'fiber' is very scarce so that its not possible for the reviewer to judge whether the use of Reissner beam model can be justified. Specifically,

(1-1) For an 'average' fiber, what is the diameter of the Kevlar core and the thickness of the PVA sheath? The authors described an overall diameter of 50 nm, their ratios to be 1:5 based on mass ratio, and their moduli to be ~960 MPa. But I question about these assumptions based on my points (1-2) and (1-3)

Our Response: The size and other nanoscale features of the fibrils were obtained from direct SEM imaging (see Supplementary Fig. 2 for example).

(1-2) Mechanical property of PVA sheath changes drastically with the water content inside of the PVA. Was the water content controlled? If so, how?

Our Response: We agree that the properties of PVA sheath could change significantly with the water content in PVA. This issue is important for the mechanics of ANF-PVA *hydrogels* reported in our previously study (ref. 1).

However, in this work, we focus on composite nanofiber *aerogels* processed with critical point drying where little residual solvent or water were present in CNAs. Therefore, water content is unlikely to be an important factor influencing the properties of PVA here.

(1-3) One way to justify the use of Reissner beam may be an experimental tensile test of 'a model core/shell fiber'. Here, my biggest concern is that we do not know the origin

of the seemingly high stretchability/compressibility of the aerogel. While Kevlar is supposed to fail at low strain, is it only PVA that sustain the overall structure of the fibers? If so, when do we see such failure?

Our response: The reviewer has raised several important questions. To answer them, we want to point out that:

(i) Due to their nanoscale diameters, it is much easier for the fibrils to accommodate imposed deformation by bending, rather than direct stretching. In addition, the deformable crosslinkers can also absorb a lot of deformation energy of the network. Indeed, as shown in the newly added Supplementary Fig. 10a below, the elastic energy associated with the stretching of fibrils only accounts for ~10% of the total energy stored in the deformed network. Another way to look at this is to examine the average stretching strain among fibrils during the deformation process. As shown in Supplementary Fig. 10b below, the average stretching experienced by each fibril is less than 2% even when the imposed macroscopic strain is above 20%.

(ii) A recent study has reported that Kevlar can sustain a tensile strain of ~4.5% (ref. 2) without fracture, which is clearly above the strain level (<2%, as pointed out above) experienced by the fibrils in CNAs. Therefore, we believe it is reasonable not to consider the fracture of individual fibrils in our model but, instead, focus on the force-induced breakage of crosslinkers.

Our modification to the manuscript: We added Supplementary Fig. 10 to show the role of fibril stretching the response of CNAs against deformation.

Supplementary Fig. 10 **The role of fibril stretching in the simulated tensile response of CNAs.** **a,** Comparison of different energies stored and dissipated in the deformed network. **b,** The average stretching strain of fibrils during the deformation process. These results show that the response of CNAs is mostly governed by the bending, enforced deformation and breakage of crosslinkers at nodal points, rather than the stretching and fracture of individual fibrils.

Comment #2: Related with the point (1), I suggest that a multiple 'hysteresis' strain cycles must be performed to supplement Figures 2a,2e, and 3g.

Our response: Following the reviewer’s suggestion, we have carried out hysteresis tests and simulations of CNAs, refer to Supplementary Fig. 14 and 15 appended below. Interestingly, by choosing the same set of parameters as those adopted in Fig. 3, good agreement between the measured hysteresis curve and our simulation results has also been achieved (see Supplementary Fig. 15).

Our modification to the manuscript: On page 10, paragraph 1, we added “The high energy dissipation capability of CNAs can also be seen from their hysteresis under cyclic tensile loading (**Supplementary Fig. 14 and 15**).”

Supplementary Fig. 14 **Multiple hysteresis strain cycles on isotropic CNAs.** Response of CNA-91 (a), CNA-87 (b), CNA-83 (c), and CNA-76 (d) under three cycles of tensile deformation, with an imposed strain of 2, 5, and 10% respectively.

Supplementary Fig. 15 **Comparison between the simulated and measured hysteresis response of CNA-76.** Simulation parameters adopted here are the same as those used in Fig. 3.

Comment #3: The description about the simulation model is also thin in the current manuscript. What does the cartoon in Figure 3d mean? I see springs and an indication that they can rotate. But what does that really mean, especially in terms of the overall network structure (Fig 3e)? And how were the "average connectivity" shown in Figure S10 calculated for all stress/strain datapoints? How does 'buckling', 'twisting', and 'stretching' taken into account in the Reissner beam model that the authors used? The explanations given in the current manuscript is not enough for readers to figure out the boundary conditions that was actually used in the manuscript; especially in terms of their relations with the actual experimental conditions.

Our response: We thank the Reviewer for bringing these important issues to our attention. Regarding the details of our model and simulation setup, we want to point out that:

(i) Every pair of fibers welded at the same node were assumed to be connected by a linear and a rotational spring, which effectively restrain their relation separation and rotation. Therefore, if a node contains three crosslinked fibrils, 3 pairs of linear and rotational springs (as shown in Fig. 3d) will be introduced to represent the interconnection among them.

Our modification to the manuscript: On page 22, we added “If a welded node contains three crosslinked fibrils, three pairs of linear and rotational springs will be introduced to represent the interconnection among them.”

(ii) Realistic 3d networks were generated from a perfect FCC lattice with dimension $6\sqrt{2}l_c \times 6\sqrt{2}l_c \times 6\sqrt{2}l_c$ where l_c is the distance between two neighboring nodes.

The network connectivity was then altered by randomly removing fibrillar segments connecting nodes until the desired connectivity was reached. At the same time, the value of l_c was chosen to ensure the fibril density equals to that in our experiment. Finally, periodic boundary conditions were enforced on each side of the simulation box. During simulation, the bottom side of the box was assumed to be fixed when the top side was forced to move in the vertical direction according to how the tensile or compression strain was applied to the material in our experiments. Finally, the average connectivity shown in Fig. S10 (now Fig. S11 in the revised submission) represents the average number of fibrils connected to each node. Due to successive breakage of crosslinkers (i.e. more fibrils are expected to detach/break from their welding points), this average connectivity will decrease with the increasing imposed strain, as illustrated in Fig. S12.

Our modification to the manuscript: On page 14, paragraph 3, the following descriptions were added:

“3D fibrillar networks interconnected by breakable crosslinkers were generated *in silico* from FCC lattice (with dimension $6\sqrt{2}l_c \times 6\sqrt{2}l_c \times 6\sqrt{2}l_c$ where l_c is the distance between two neighboring nodes) as proposed by Broedersz *et al.*³⁰. Notice that perfect FCC lattice networks have an initial coordination number $z = 12$, placing them well above the central-force (CF) isostatic threshold connectivity ($z_{cf} = 6$) according to the Maxwell rigidity theory. In contrast, most previous models for stiff-fiber networks have a maximum coordination number 4. Here, we explore the effects of network connectivity by randomly removing fibrillar segments between nodes until the desired connectivity was reached. At the same time, the value of l_c was chosen to ensure the fiber density equals to that in our experiment.”

And in Supplementary Fig. 11, we added “**Representative 3D networks with distinct average nodal connectivity (\bar{z}) defined as the average number of fibers connected to each node.**”

On page 15, paragraph 1, the boundary description was added as “Finally, periodic boundary conditions were enforced on each side of the simulation box. During simulation, the bottom side of the box was assumed to be fixed when the top side was forced to move in the vertical direction according to how the tensile or compression strain was applied to the material in our experiments and the reaction force was recorded, allowing us to construct the stress-strain curve.”

(iii) Lastly, we want to emphasize that geometric nonlinear effects (due to, for example, rotation, bending and stretching) and twisting of beams have all been taken into account in the Reissner theory (ref. 4). Specifically, in addition to taking into account large deformation and rotation of the beam, a spin matrix is also introduced to describe the twisting of its cross-section. Therefore, this theory can capture the buckled and twisted shape of individual fibrils. To demonstrate this, a snapshot of the several deformed fibrils (from our simulations) is shown below where the buckled shape of a fibril can clearly be seen. In addition, details and examples of using Reissner theory to study the buckling and twisting of beams can also be found in, for example, ref. 3 and ref. 4 provided at the end of our response.

Snapshot showing the deformed shapes of several interconnected fibrils from our simulations. It can be seen that the red fibril assumes a buckled configuration due to the compressive load it sustains.

Our modification to the manuscript: On page 15, paragraph 1, we added “Finite element method was then used to capture the deformation responses of networks where each fibril was modelled as a three-dimensional Reissner beam³⁶ that can undergo large rotation, stretching, bending, and twisting.”

Comment #4:

(4-1) "The average pore sizes of CNAs range from 140 nm to 1463 nm..." ==> How were these values quantified?

Our response: As stated in methods part: the pore size distribution of CNAs with different densities was measured by mercury intrusion porosimeter (MIP, AutoPore IV 9600) with pressure ranging from 0.1 to 6.1×10^4 psi.

(4-2) "For anisotropic CNAs with a porosity of 48.7%" ==> Porosity seems to be quite low. Can this be classified as 'aerogel'?

Our response: We have modified the text accordingly to avoid potential controversy.

Our modification to the manuscript: On page 7, paragraph 2, we changed “anisotropic aerogels” to “anisotropic porous composites”.

References

1. Xu, L. Z., Zhao, X. L., Xu, C. L. & Kotov, N. A. Water-rich biomimetic composites with abiotic self-organizing nanofiber network. *Adv. Mater.* **30**, 1703343 (2018).
2. Trexler, M. M., Hoffman, C., Smith, D. A., Montalbano, T. J., Yeager, M. P., Trigg, D., Nimer, S., Calderón-Colón, X., Peitsch, C. & Xia, Z. Y. Synthesis and mechanical properties of para-Aramid nanofibers. *J. Polym. Sci. B: Polym. Phys.* **57**, 563-573 (2019).
3. Crisfield, M. A. . Non-linear finite element analysis of solids and structures. Vol. 2: Advanced topics. (John Wiley & Sons, Inc., NewYork, 1997).
4. Jelenic, G. &Crisfield, M. A. Geometrically exact 3d beam theory: implementation of a strain-invariant finite element for statics and dynamics. *Comput. Method. Appl. M.* **171(1-2)**,141–171(1999).

Reviewers' Comments:

Reviewer #1:

Remarks to the Author:

I looked at the responses and the revised manuscript. The authors have addressed all my questions and comments, and I think it can be published.

Reviewer #2:

Remarks to the Author:

The authors' replies make sense and now I am convinced with the work in scientific sense. For most readers who read the manuscript and not-as-much the Supporting Information may not find it difficult to make sense. I suggest that the authors blend Supplementary Figure S10 and the last snapshot in the author's reply (the one with the red fibril) into main Figure 3. The authors want to add more explanations about it in the text. The authors can find good arguments written in the reply to the reviewers, so I believe that it is not too much work for them.

Comments from Reviewer #1

Summary: I looked at the responses and the revised manuscript. The authors have addressed all my questions and comments, and I think it can be published..

Our Response: We thank the Reviewer for the support for the publication of our work in *Nature Communications*.

Comments from Reviewer #2

Summary: The authors' replies make sense and now I am convinced with the work in scientific sense. For most readers who read the manuscript and not-as-much the Supporting Information may not find it difficult to make sense. I suggest that the authors blend Supplementary Figure S10 and the last snapshot in the author's reply (the one with the red fibril) into main Figure 3. The authors want to add more explanations about it in the text. The authors can find good arguments written in the reply to the reviewers, so I believe that it is not too much work for them.

Our Response: We thank the Reviewer for the support for the publication of our work in *Nature Communications*. We have further revised the manuscript to address the Reviewer's comment.

Our modification to the manuscript: On page 9, paragraph 1, we added “The elongation of CNAs is mostly governed by the bending, enforced deformation and breakage of crosslinkers at nodal points, rather than the stretching and fracture of individual fibrils (**Supplementary Fig. 10**).”